# Toward High-Quality Adult Online Learning: A Systematic Review of Empirical Studies

Yefeng Lu, Xiaocui Hong * and Longhai Xiao

College of Education, Zhejiang University, Hangzhou 310058, China; 12103004@zju.edu.cn (Y.L.); zdlonghai@zju.edu.cn (L.X.)
* Correspondence: 11903005@zju.edu.cn

**Abstract:** Adult education is a key policy to achieve the sustainable development goals. Large-scale open online courses are gradually increasing with the continued spread of COVID-19 all around the world, which has also attracted more and more adults to participate in such courses. However, despite the fact that the research on adult online learning is abundant, there is still a lack of systematic summaries that can guide the design and selection of course content and instructional methods. Therefore, the purpose of this study is to systematically examine the factors and related strategies that influence adult online learning, and to some extent also provide directions for future research. Using a systematic literature review, with the help of literature visualization software CiteSpace, this study summarized and analyzed 124 SSCI literature of empirical studies. The findings show that although some conclusions of adult online learning research are controversial, there is still some consensus, which is worthy of our attention. First, adult learners have time constraints and more responsibilities, hence life oriented, structured, and flexible online courses are more suitable for them. Secondly, adult learners have less scholastic aptitude and are less ICT skilled than normal students, so preparatory learning is necessary. Finally, in terms of an online instructional approach, integrated online discussions are recommended, as adults are prone to inefficient and superficial discussions in open discussions. This study contributes to theory and practice by expanding the systematic understanding of online learning for adults.

**Keywords:** adult online learning; andragogy; online instruction; systematic literature review

## 1. Introduction

Adult education is a key policy to address the challenges of employment and re-employment, digital transformation, globalization, aging, climate crisis, and as an important measure to achieve the Sustainable Development Goals [1]. Over the past few decades, a lack of learning opportunities and resources has prevented large numbers of adults from participating in necessary education and training. Fortunately, in recent years, with the development of ICT, online learning has in some ways dramatically removed the boundaries of education and has enabled more and more people to access educational opportunities and resources [2]. Flexibility, accessibility, and not being limited by physical time and space make online learning an important means of bridging formal and informal learning and facilitating the realization of learners' vision of lifelong education [3]. This was especially pertinent during the lockdown period of the COVID-19 pandemic, with educational institutions in 194 countries closing down [4]; it has been proposed that online learning become an important alternative form of learning for higher education and vocational training [5–7]. In addition, digital technology serves not only as a tool for learning, but also as a cultural medium for sharing ideas, and it can help adult students develop critical thinking about topics such as climate change, inequality, political conflict, and other issues related to sustainable development [8]. However, there has been a lingering paradox in adult online learning. On the one hand, adults need to take on more family and work

responsibilities. Limited by time and space, flexibility and accessibility are the main reasons why they choose online learning [9,10]. On the other hand, it has to be admitted that online learning also brings a large number of online dropouts [11,12], which in turn limits its transformative potential.

Adult learners themselves indeed have many limitations (e.g., lack of time, work pressure, etc.), but we should also reflect whether online learning courses are designed according to the needs of sustainable development and characteristics of adult learners. Related studies have reported some problems in online course design, involving learning materials [13], interactions [14,15], and evaluation [16].

Recently, some researchers have paid attention to learning differences between adult learners and ordinary students, and have proposed corresponding instructional designs. For example, Cercone (2008) systematically analyzes the characteristics of adult learners and proposes 13 instructional design recommendations [17]. Ekmekci (2013) proposes a curriculum structure and a learning intervention model based on experiential learning, adult learning principles, case-based, and problem-based learning approach, and peer review [18]. Arghode et al. (2017) compares theories related to adult learning and distinguishes learning processes under different theoretical directions [19]. Despite the value of these studies in guiding course design and adult online learning program practice, most of them are top-down deductions, which means that some of the conclusions may be unproven, and the complex context of practice and the diverse influencing factors may even be ignored. In this light, a bottom-up summary based on empirical research is necessary. In addition, the emerging empirical findings and the divergent themes also need to be summarized and reviewed, otherwise, it will be difficult to guide adult online learning practices systematically. Hence, the aims of this paper are as follows:

(1) Examine the factors and related strategies that influence adult online learning systematically, and

(2) Provide directions for future research based on the status, issues, and trends of existing adult online learning research

## 2. Research Method

To achieve the purpose of the study, we performed a systematic review with the help of visual analysis software CiteSpace.5.8. R2 (developed by Chaomei Chen, http://cluster.cis.drexel.edu/~cchen/citespace/, accessed on 26 December 2021). A systematic literature review emphasizes a rigorous and transparent literature search and selection process that can avoid researcher bias in a literature review to a certain extent [20,21]. As for CiteSpace, it is a software developed to meet the need for visual analysis of literature [22], which can execute functions such as co-citation analysis, keyword co-occurrence analysis, cluster analysis, burstiness analysis, and social network analysis of literature [23]. In this study, we used the systematic literature review method, following PRISMA guidelines [24] to search the literature, identify inclusion and exclusion criteria, evaluate the literature, and extract and analyze important findings of the literature, while CiteSpace was mainly used for bibliometric statistics to more comprehensively help to understand research trends and correlations between different research topics, etc. The research framework is shown in Figure 1.

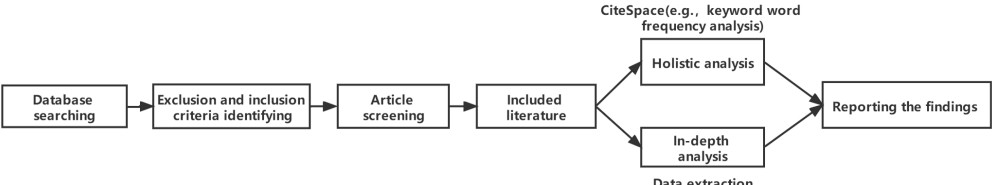

**Figure 1.** Process of the systematic review.

## 2.1. Database Searching

To ensure the quality of the literature and the credibility of the research results, we chose articles published in SSCI index journals in Web of Science (WoS) database as the source of the literature, which covers the majority of authorized journals in the field of social sciences all around the world and has a relatively rigorous peer review scheme. In terms of search terms, considering the variety of expressions for adult online learning, we list the potentially relevant keywords in advance (as shown in Table 1). After that, we combined the terms for article searching: ("Adult* learning" OR "Senior learning" OR "Elderly learning") AND ("mobile device*" OR "technology*" OR "smartphone*" OR . . . OR "ICT*"). In addition, a time span of between 2005 and 2021 was set, as the year 2005 is generally considered to be an important time node when Information and Communication Technology (ICT) became widely accessible [25]. A total of 578 articles were found in WoS in the initial search.

**Table 1.** Search parameters and specifications.

| Theme Specification | Potential Key Subject Terms | Other Filter Parameters |
| --- | --- | --- |
| Adult learning, senior learning, elderly learning | mobile device, technology, smartphone, tablet, social media, social network, Internet, online, app, We-Media, Moocs, open resource, online community, ICT | Since 2005; literature type "articles" |

## 2.2. Identifying Exclusion and Inclusion Criteria

To remove the redundant or duplicated articles that did not meet the criteria, inclusion and exclusion criteria were identified as followings: (1) the research topic must focus on "adult online learning"; (2) the research methods must conform to the norms of empirical research; (3) the research sample must be adult students. According to Richardson and King's definition [26], "adult students" refer to students who returned to or re-entered their post-secondary education at an age of 22 or over, or enrolled on less than a full-time basis. This definition excludes research with samples of undergraduate and graduate students; and (4) the experimental context must be fully clarified, otherwise, we could not judge whether the technology composed the online learning environment or the technology was just used as an assistant to traditional face-to-face instructions.

## 2.3. Article Screening

Based on these four criteria, we carried out further article screening according to the following steps (see Figure 2). In the first round, we perform an initial screening of the articles based on the titles and abstracts provided by WoS to determine if the research topics and research methods meet the criteria. If met, we would attempt to download the literature. The literature that was accessible went to the second round of screening, focusing on reading and screening the sections on research context and sample information to determine exclusion or inclusion. The remaining articles that could not be judged at that moment entered the third round. In this round, we read the full text to find out if there were clues to meet the screening criteria. Those articles for which we could not find adequate supporting information would be excluded.

## 2.4. Data Extraction

Data extraction was used to help us draw key information from the literature more easily. We coded the literature for the following: researcher, publication date, research method, sample information, and main findings of the study (see Table 2).

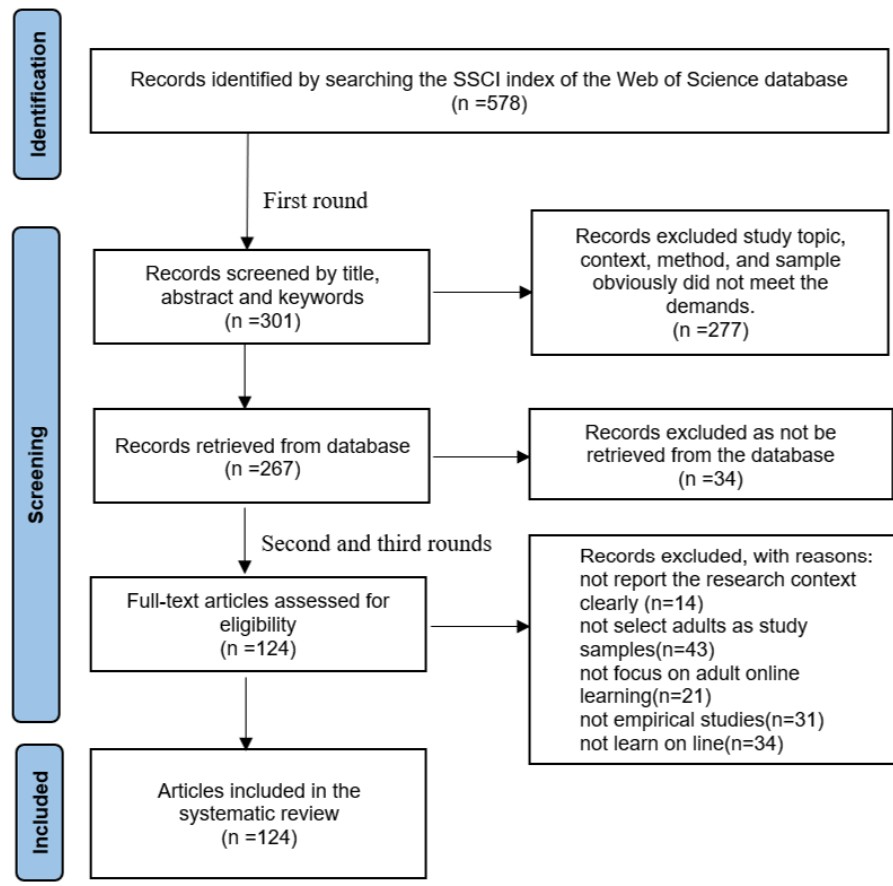

**Figure 2.** The process of literature search and selection.

**Table 2.** Article coding items.

| Code Items | Introduction |
|---|---|
| Researcher and publication date | For simplicity, we put the two items together (e.g., Park et al., 2009). |
| Research method | Research methods were coded as quantitative, qualitative, or mixed methods |
| Sample information | We extracted as much information about the sample as possible, including the age, number, and work background of the participants (if mentioned). |
| Main findings | We would summarize the empirical findings in the paper, but untested exploratory models or theoretical frameworks based on the evidence are not included. |

*2.5. Literature Analysis*

In the literature analysis phase, we conducted a holistic analysis and an in-depth analysis. The aim in the holistic analysis was to capture important themes and changing trends in adult online learning research, and thus we selected three analysis items: published year analysis, burstiness words analysis, and keyword co-occurrence analysis. Among them, published year analysis can reveal the changes in the number of publications over time, and burstiness words analysis can reflect the important themes or research hotspots that have appeared in past studies, both of which can help us understand the changing trends of the research theme of adult online learning. Keyword co-occurrence analysis is used to reveal associations between themes as well as to identify important influencing factors. For example, the keyword "usage" often co-occurs with keywords such as "gender", "attitude", and "community", which suggests that there are associations between these topics. More importantly, these keywords co-occurring with "usage" are likely to be potentially important influencing factors. Through the three analysis items

mentioned above, we can effectively capture the important themes and avoid possible thematic omissions or deviations in the in-depth analysis.

In contrast to the holistic analysis, the in-depth analysis focused on analyzing, comparing, and categorizing the coded results. We did two works in this session, one is to classify the examined influences, and the other is to organize the consistent or mutually supporting research findings. To ensure the reliability of the in-depth analysis, the two authors (first and second author) independently collated and analyzed the coding results. In case of large conflicting conclusions of the articles, we will carefully compare and discuss the research background, sample information, and research methods to understand the potential reasons.

## 3. Results

### 3.1. Holistic Analysis of the Included Literature

#### 3.1.1. Published Year Analysis

Analyzing the trend of published articles over the years can help us to grasp the overall dynamics of the research. We presented the annual distribution of these 124 papers (see Figure 3). Since 2015, there has been a sudden increase in research on adult online learning. This may be due to the popularity of online learning, which has attracted widespread attention from scholars.

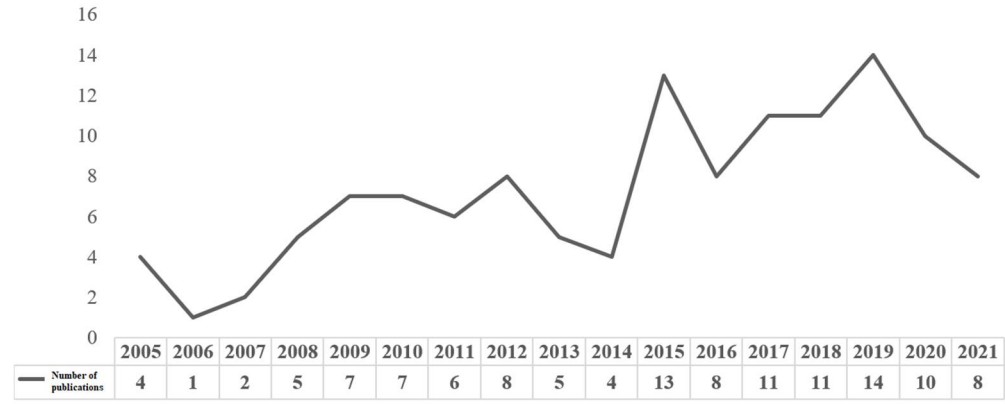

**Figure 3.** The annual distribution of the publication.

#### 3.1.2. Burstiness Words Analysis

Burstiness words refer to keywords that appear suddenly within a certain period of time [27]. By analyzing the years in which these keywords appeared, we can understand the historical threads and future research trends in the field of adult online learning. Figure 4 displays the top 25 keywords with a duration of at least two years of strength ranking. Among them, attitude, cognitive, development, community, technology acceptance, and learner lasted for more than 4 years, which reflected the focus of the adult online learning research field.

#### 3.1.3. Keyword Co-Occurrence Analysis

Keywords are the core summary of an article, which, in a sense, reflects the theme and main content of the article. The keywords given in one article are necessarily related to each other in some way, and this connection can be measured by the intensity of keyword co-occurrence. It is generally believed that the higher the frequency of two or more keywords appearing together, the stronger the connection between them [28]. CiteSpace provides a function called Betweenness Centrality to describe the strength. In short, if a keyword always co-occurs with other different keywords, it means that we would meet this keyword even if we discuss other related topics. Accordingly, the higher the value of Betweenness Centrality that this keyword shows, the more important the status may be. As depicted in Figure 5, the larger nodes and labels imply larger values of Betweenness Centrality.

Additionally, keywords such as usage, teacher, technology, participation, education, model, performance, technology acceptance, etc., show high values of Betweenness Centrality in the figure. Through the links in the figure, other keywords that co-occurred with a certain keyword can be found easily. For example, the keyword "technology acceptance" was highly correlated with keywords "usage, efficiency, satisfaction, intention, etc.". To find the relevance of different articles, we listed the top 10 keywords with Betweenness Centrality values and five main keywords that co-occurred with them, respectively (see Table 3). Additionally, it came out that the main foci of current research on adult online learning were individual-level influences (e.g., age, efficacy, and motivation, etc.), the human-technology interaction (e.g., technology acceptance, internet self-efficacy, etc.), and the course experience (community, support, etc.).

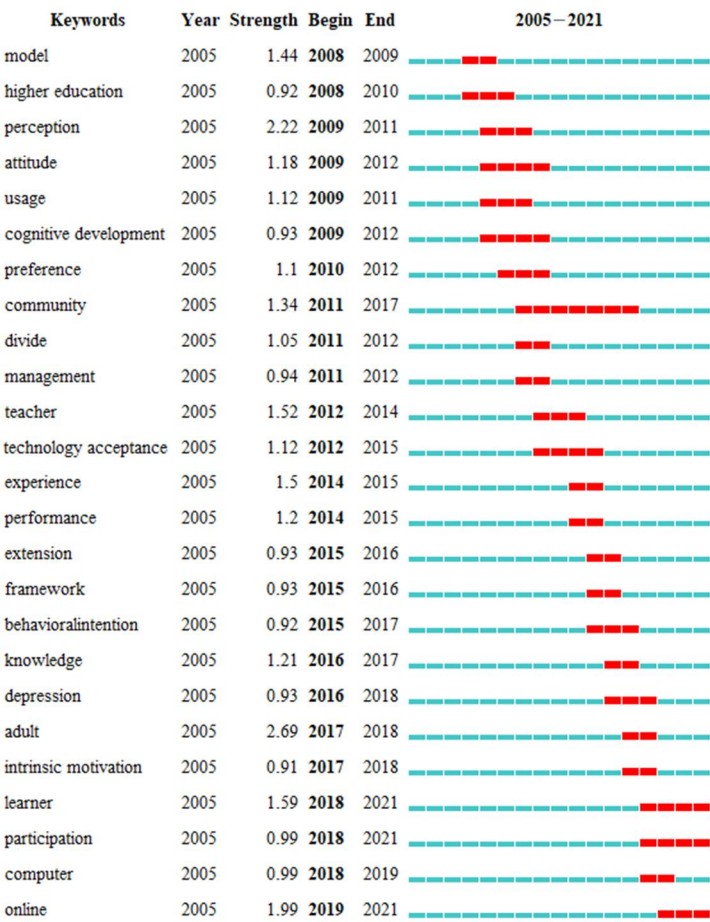

**Figure 4.** Analysis of TOP 25 burstiness keywords.

### 3.2. An In-depth Analysis of the Included Literature

Through the analysis of the coding results, combined with the findings from the previous keyword co-occurrence analysis, the research topics and contents were further classified. In general, articles focused on exploring the influencing factors, including individual characteristics and internal and external factors, which had a certain cross effect on adult online learning. Besides, some articles also examined articles that examined related strategies, such as comparing the effects of different instructional strategies and learning approaches on learning effectiveness. To present the findings more clearly, a preliminary topic framework was constructed (see Figure 6).

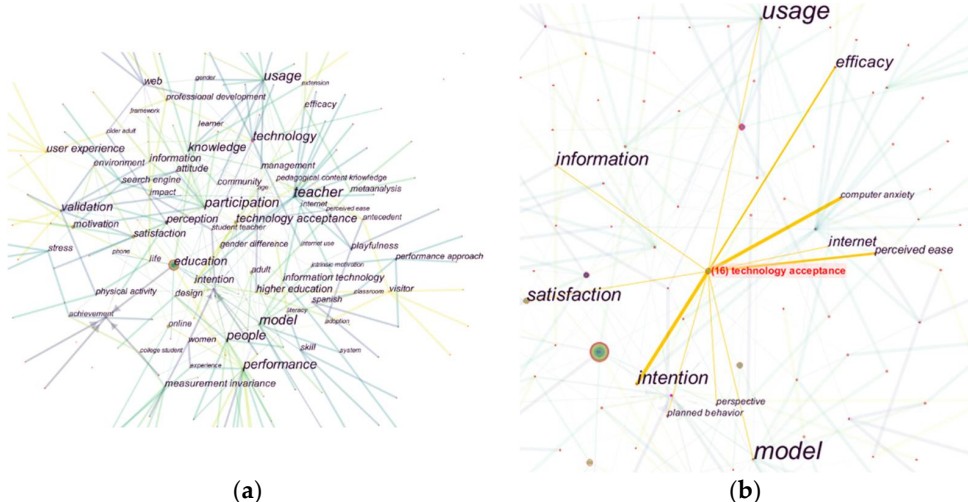

| (**a**) | (**b**) |

**Figure 5.** Keyword co-occurrence mapping. (**a**) Network of keyword links; (**b**) Co-current keywords with "technology acceptance".

**Table 3.** Top 10 keywords and top five co-occurring keywords.

| Main Keywords. | Co-Current Keywords |
| --- | --- |
| Technology | Professional development, performance, internet use, internet self-efficacy, information |
| Model | Performance, information technology, usage, technology acceptance, gender difference |
| Teacher | Playfulness, readiness, support, community, discussion |
| Participation | Community, motivation, satisfaction, efficacy, management |
| Performance | Information technology, model, competence, experience, cognitive load theory |
| Usage | Technology acceptance, perception, attitude, gender, community |
| Education | Teacher, knowledge, outcome, satisfaction, life |
| People | Adult, intention, perception, internet use, model |
| Technology acceptance | Model, satisfaction, usage, efficacy, information |
| Knowledge | User experience, participation, age, management, usage |

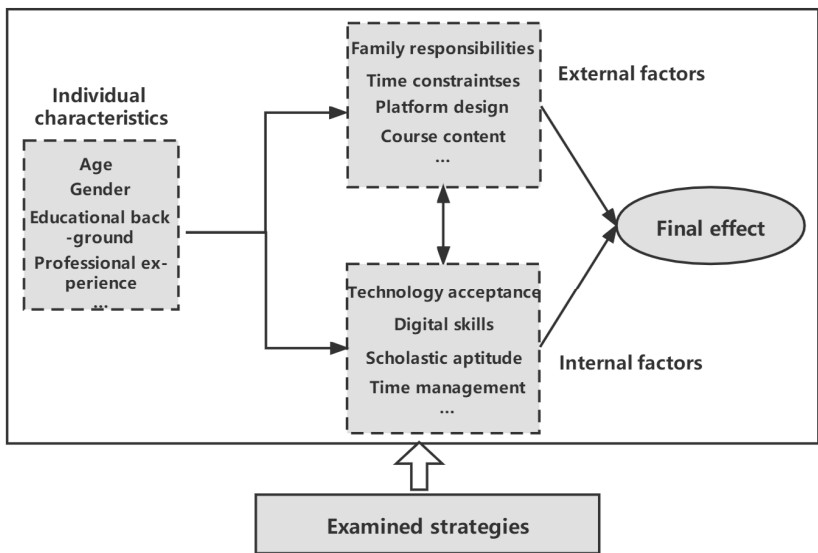

**Figure 6.** Classification of research topics.

### 3.2.1. Adults' Individual Characteristics

In terms of the influence of individual characteristics, related research findings seem conflicting. For example, Coryell and Clark (2009) have analyzed the influence of gender on adult online learning [29]. They suggest that online learning might be an exhausting and difficult learning process for women because of their various responsibilities as mother, wife, etc. Furthermore, many studies in the field of technology acceptance confirm that women have lower self-efficacy when dealing with technology [30,31], and ICT self-efficacy is in turn a predictor of participation in online learning [32]. By this reasoning, women should be less engaged in online learning than men. However, Diep et al.'s (2016) study shows that working learners, lower degree holders, and females are the keenest participants in online participation [33]. The reason for this conclusion is that women are more socially interactive, whereas men are more information-based. Thus, females would be more active in online learning. Contradictory findings also appear when focusing on the impact of age on learning. A large amount of research claims that the elderly are more absorbed in online learning and are willing to devote more time to learning [34–36]. However, Park and Choi (2009) found no difference in dropout rates for online learning between ages [37], which indicates that the argument that the elderly are more interested in learning is dubious. In addition, inconsistent findings also emerge in individual characteristics such as education levels, career experiences, race, and so on. Overall, the effect of individual characteristics remains controversial. Some researchers believe that the effect of individual characteristics may be weak [37], and are not directly predictive [38]; accordingly, they may act as a moderating factor. For example, Xiong and Zuo (2019) found that the main purpose for young people to learn online is to solve problems, while the purpose for the elderly is to acquire knowledge (e.g., health knowledge, fishing tips, etc.) [39]. Although age does not play a direct role in adult online learning, it can have an impact on the effectiveness of the course content itself. Therefore, different course content needs or preferences of learner groups of varied ages need to be taken into account.

### 3.2.2. External Factors Affecting Adult Online Learning

External factors affecting adult online learning focus on two aspects: physical constraints (work chores, family responsibilities, time constraints, etc.) and course-related factors. In terms of physical constraints, the consensus is that adults need to deal with more chores or tasks from their family, work, etc., and thus they are prone to online absenteeism or low efficiency. Choi and Park (2018) revealed that physical constraint variables directly or indirectly influence adult online dropout by affecting course content, satisfaction, and GPA [10]. Adult learners who endure a variety of physical constraints struggle to persist with online learning if they do not obtain adequate support and encouragement. Park and Choi (2009) proved the significance of family and organizational support for sustainable learning [37]. Support by families and organizations has a positive impact on adults' online learning [40–42]. They ultimately affect learners' intentions to continue learning by influencing mediating variables of perceived usefulness [40,43]. It can be imagined that when seeing family members and colleagues backing out of online learning and arguing that it is useless, adult learners' value judgments towards online learning are likely to be disturbed. Additionally, this would eventually lead to dropouts. In addition, adults' physical constraints also influence their course preferences to some extent. Adult learners prefer courses that are flexible and relevant to their occupations [32,37]. It is easy to understand that flexibility allows adult learners to set their own pace, which can avoid scheduling conflicts between learning and work. As for the relevance, this is also in line with Knowles' theoretical assumption of Andragogy that adult learning is not about preparing for future life, but about solving problems in the present [44].

Course-related factors or variables examined include: credibility, transparency, and productivity at the institutional management level; perceived ease of technology use, relative technical advantages, technical complexity, compatibility at the human-technology interaction level; and classroom interactivity, flexibility, evaluation, satisfaction, and flow experience at the instruc-

tional level [15,45–48]. Researchers usually explore the impact of relevant influencing factors on satisfaction. For example, Ilgaz and Gulbahar (2015) found that problems with technology and evaluation were the main reasons affecting satisfaction [16]. Lee and Choi's research (2013) showed that learning strategies, flow experiences, and internal locus of control had a direct or indirect effect on satisfaction [46]. In a more specific qualitative survey, Ruey (2010) found that the lack of evaluation standardization would cause dissatisfaction among adult learners, for it was difficult to distinguish between learners who worked hard and those who did not [47]. Ge's study (2011) on adult online English learning proved that learners with higher writing skills, when mixed with learners with lower writing skills, tend to become frustrated. They felt there was nothing to learn from their peers [49]. In a short, although satisfaction is an important variable, there are so many variables that affect satisfaction, and the framework to describe how course-related factors affect satisfaction has not yet been established.

### 3.2.3. Internal Factors Affecting Adult Online Learning

Among the internal factors influencing online learning, factors related to scholastic aptitude (including self-regulated learning, learning strategies, achieving goals, etc.) and factors related to technology use (including self-efficacy, perceived ease of use, perceived usefulness, social influence, etc.) are the most mentioned factors. First, the use of technology is fundamental to online learning. Once adults refuse to accept the technology of online learning (e.g., learning systems or mobile devices), they will drop out of online learning directly. Thus, technology acceptance is one of the most frequent research topics in this field. Relevant research mainly relies on the Unified Theory of Acceptance and Use of Technology (UTAUT), Technology Acceptance Model (TAM), and other models derived from them [42,43,48,50]. Adult learners, as digital immigrants, are likely to have some technical barriers when using ICT for learning, especially older adults. Not only do they have problems using technology, but they also show distrust of technology, including fear of information leakage and Internet fraud [48]. Eynon and Malmberg's research (2021) shows that while online learning offers convenience to learners, it excludes adult learners who lack digital skills, which means that this group does not benefit from the development of technology [51]. Among all the variables affecting technology acceptance, perceived ease of use, perceived usefulness, and factors related to perceived usefulness, such as job-fit, enjoyment, and performance expectancy, have been confirmed to have a direct impact on learners' attitudes and behavioral intention to use technology [48,50,52].

Technology use is indeed the main influencing factor that adult learners encounter at the beginning of their exposure to online learning. However, if an individual has fully mastered the use of technology or initially has high technological literacy, the issue of technological barriers will become less prominent. In this case, factors related to scholastic aptitude become pivotal. Choi and Park (2018) suggest that scholastic aptitude affects GPA and indirectly causes adult learners to drop out of online learning [10]. Lai (2011) investigated the online learning situation of civil servants in Taiwan, and it was found that self-directed learning readiness and network literacy were significant predictors in predicting the online learning effectiveness of civil servants [53]. In addition, intrinsic motivation, metacognition, and self-regulated learning, learning strategies, core self-evaluation, and self-efficacy, etc., have also been mentioned [53–55]. In these related aspects regarding scholastic aptitude, adult learners are inferior to ordinary students. Boelens et al.'s interviews (2018) with teachers revealed that teachers generally felt that adult students are not as good at monitoring their learning as college students [56]. Hood et al.'s (2015) survey of Mooc learners also shows that learners studying towards a higher education degree are significantly better at self-regulating their learning than low-education or employed adult learners [57]. Therefore, particular training and support for adult learners in scholastic aptitude may be critical for improving their effectiveness of online learning. Table 4 provides a summary of the relevant influencing factors mentioned above.

**Table 4.** Summary of influencing factors on adult online learning.

| Type of Influencing Factors | Main Factors Mentioned | References |
|---|---|---|
| Individual characteristics | Prior experience | [33,57,58] |
| | Gender | [33,34,59] |
| | Age | [36,37,58] |
| | Educational level | [33,37,57] |
| | Race | [60] |
| External factors | Physical constraints | [10,16,37] |
| | Family and organizational support | [37,40] |
| | Learning support | [61] |
| | Interaction | [15,45] |
| | Evaluation | [16,47] |
| | Course format, type, structure, etc. | [38,62,63] |
| | Technology characteristics | [32,64] |
| | Institutional management | [45] |
| Internal factors | Digital skills | [51,53,65] |
| | Scholastic aptitude | [10] |
| | Locus of control | [46] |
| | Achievement goals | [66] |
| | Core self-evaluation | [55] |
| | Self-regulation skills | [54,57,67] |
| | Learning styles/preferences | [68] |
| | Anxiety | [48,69] |
| | Perceived usefulness | [40,43,48] |
| | Perceived ease of use | [43,48,52] |
| | Self-efficacy | [43,50] |

### 3.2.4. Examined Online Learning Strategies for Adults

In order to improve learning effects, some researchers also compared and analyzed different course types, course content, learning assessment, and interactions. In terms of course types, Ge (2012) compared the single cyber asynchronous learning approach with the blended cyber learning approach in English distance education [63]. The results showed that although both approaches improve adult learners' final score, the blended approach could bring a significantly better result in their English study than the single cyber asynchronous approach. Synchronous online learning focuses more on classroom interaction, while asynchronous classes focus on deep cognitive engagement, which is of great guiding significance to design online learning. In addition, after-school support, such as e-mail reminders and communication, helps improve course satisfaction for adult learners [15]. Thus, a blended form of online learning with face-to-face interventions is suggested [70]. Regrading course content, adult learners prefer content that is closely related to their work or life [37,52], such as professional development, emotional problems, health problems, etc. [71]. Besides the cognitive characteristics, adult learners require the course content to be structured. A clear syllabus and learning objectives should be provided at the beginning of each week so that adult learners can be facilitated to connect the content with the objectives [72]. Ke and Xie (2009) argued that in a structured course in which reading materials, lectures (including PowerPoint slides, learning videos, and lecturer notes), and assignments (including evaluation rubrics and sample answers) are

clearly presented on an online platform, adult students have little difficulty in learning and the final completed assignments are well standardized [38]. As for assessment, the interactive assessment approach seems to have a certain impact effect on adult learners. In Ge's (2011) research on adult English online learning, peer review was found to be helpful to adult learners with low writing ability [49]. Ruey (2010) also supports this view by finding that evaluating other people's views and ideas often helps adult learners solve the same issues from different perspectives [47]. However, there seemed to be some controversial views on interaction. Some articles reported that interaction is beneficial for increasing trust and eliminating anxiety [29,73,74], but others reported that adult online interactions were largely uninvolved in higher-order thinking and were often one-way, individualistic, and superficial [14,75]. Furthermore, the continuously emerging statements and diverse topics may prevent any topic from being discussed in depth [47]. Thus, Ke and Xie (2009) suggest that the close-ended and open-ended discussion should be combined to maintain the openness of the discussion without departing from the topic [38]. Overall, many strategies have been examined by relevant scholars and these works are undoubtedly very meaningful. However, exactly how these are related to internal and external factors is still lacking exploration.

In order to better understand how to support adult online learning, the relevant research conclusions have been integrated to build an integration framework of factors affecting the quality of adult online learning, as shown in Figure 7 below. In this framework, we distinguished two phases of course learning: early and late course learning. In the early stages of learning, life issues and professional development drive the adult learner's need to learn. Without this intrinsic need, of course, online learning could not happen. When adults begin to participate in online learning, many issues such as technological barriers and physical constrains need to be overcome. If these problems are not solved well, it is difficult for them to continue learning. Therefore, adult technology acceptance, as well as family and organizational support, needs to be a key focus in the early stages of learning. In the later stages of learning, when adults have fully adapted to the online learning format, individual scholastic aptitude, course design, and interaction have a critical impact on the learning outcome and course experience. Superficial discussions and poor course content structure need to be avoided.

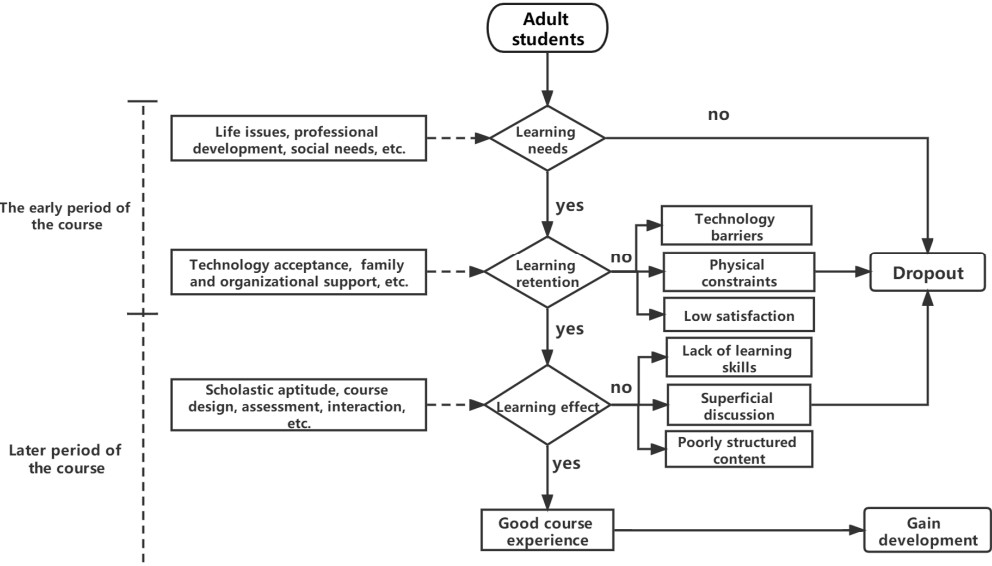

**Figure 7.** An integration model of factors affecting the quality of adult online learning.

## 4. Discussion

The number of articles on adult online learning is increasing and the research topics are becoming so divergent that we must systematically sort, compare, and examine them to

properly use the findings to guide the implementation of adult online learning programs. Using a systematic literature review with the help of the literature visualization and analysis tool CiteSpace, 124 SSCI empirical research papers in WoS were collected and analyzed. An overall analysis showed that the number of articles referring to adult online learning grew faster from 2014–2015, which may be attributed to the popularity of online course platforms, such as Moocs. In the analysis of keywords, attitude, cognitive development, community, technology acceptance, and learners were the keywords that have been of interest to researchers for a relatively long time. In addition, we also conducted keyword co-occurrence analysis for the top 10 keywords with Betweenness Centrality values. On this basis, an in-depth analysis of the collected articles was conducted following the individual characteristics, external factors, internal factors, and examined online learning strategies and an integration model of factors affecting the quality of adult online learning was built.

In our analysis of the factors influencing adult online learning, we found that the findings of related studies are conflicting in terms of individual characteristics, and thus we agree with Park's view that the effect of individual characteristics may be weak [37]. However, we also need to consider the association of individual characteristics with course content preference, available time, and personal experience. Among the internal and external factors, physical constraints, course-related factors, technology acceptance, and scholastic aptitude are the important influences that relevant researchers consistently identify and that course developers must pay attention to when designing online courses. In addition, there are also studies that directly examine the impact of different instructional strategies. Undoubtedly, these studies are valuable. However, the link between influencing factors and specific strategies remains obscure. Future research is expected to bridge the two divides.

## 5. Conclusions

As a summary, the initial purpose of this study is to provide a more comprehensive understanding of adult online learning through literature summaries and to provide effective guidelines for practice. Although the results of some studies on the characteristics, influencing factors, and effective strategies of adult online learning are controversial, there is still some consensus that can guide practice to some extent:

(a) On the choice of learning topics, the special characteristics of life stages determine that adult learning is a difficult trade-off between payoffs and rewards. Not feeling the real benefits of online learning will increase the dropout rate. Therefore, relevant institutions should design life-based online courses, which are intended to help adult students think critically and reflectively about life issues or career problems, rather than deliver knowledge simply. Some programs' curriculum design concepts are worth learning from, such as the M.Sc. program [76]. Curriculum are seen as realistic, complex, interactive learning environments, in which adult learners are encouraged to engage in problem-based learning and inquiry-based learning. This project offers excellent experiences, providing opportunities for reflective practice, using ICT tools to facilitate learning, and developing interactive, interdisciplinary, and cross-disciplinary skills.

(b) In terms of online course content design, due to the limitations of learning time and cognitive ability, well-structured course content is more appropriate. For example, the provision of a syllabus, course modules, learning objectives, learning materials, evaluation rubrics, and sample answers. This content should be clearly presented on the online learning platform to facilitate learners to browse and learn. Some web-based course programs (although conducted in primary and secondary schools) with flexible structures are also worthwhile [77]. Their learning content is presented in the form of units that are not dependent on previous units. Thus, these units and activities can then be implemented flexibly according to the competency base and needs of the learners.

(c) In terms of instructional methods for online learning, while adult learners enjoy classroom interaction, it is often superficial and does not promote in-depth knowledge construction. It may be that integrated discussions work better than fully open discussions

(i.e., giving adults an open discussion based on experience on the one hand and providing summary statements or authoritative conclusions on the other).

(d) In terms of external support, adult learners are inferior to ordinary students in general learning skills, such as digital skills and self-regulated learning skills. It is necessary to conduct preparatory learning for learning general skills at the beginning of the courses, such as informing adult learners how to manage their time, how to obtain effective academic help, and how to use e-learning platforms, etc.

Finally, there are certain limitations to the work of our literature review, which we must point out. Because of our team's capacity and time constraints, we only selected the SSCI search database of WoS, which may have led us to miss other literature of significant value in our literature review. We also expect a researcher to conduct a literature review with a broader database of literature sources, a larger size of literature volume, and a more detailed literature analysis to fill the gaps in our work.

**Author Contributions:** Conceptualization, Y.L. and X.H.; methodology, Y.L.; software, Y.L.; validation, Y.L., X.H. and L.X.; formal analysis, X.H.; investigation, Y.L.; resources, Y.L.; data curation, Y.L.; writing—original draft preparation, Y.L.; writing—review and editing, X.H.; visualization, L.X.; supervision, L.X.; project administration, Y.L.; funding acquisition, L.X. All authors have read and agreed to the published version of the manuscript.

**Funding:** The research was funded National Office for Education Sciences Planning of China [grant number: DHA100260].

**Institutional Review Board Statement:** Not applicable.

**Informed Consent Statement:** Not applicable.

**Data Availability Statement:** Not applicable.

**Conflicts of Interest:** The authors declare no conflict of interest.

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
