# Peer review of "Toward High-Quality Adult Online Learning: A Systematic Review of Empirical Studies"

_sustainability, doi:10.3390/su14042257_

Round 1

Reviewer 1 Report

  1. The objectives of the paper can be specified in the abstract (as they are presented in the introduction). This will make the abstract sufficient and accurate to introduce the ideas presented by the authors.
  2. The authors can separate a section Conclusion to summarize and highlight the achieved results. At the moment, the conclusion is intertwined with the discussion.

Author Response

Dear Reviewers

        Your comments were very on-target, which was very helpful in improving the paper. We have revised it according to your comments, and specifically:

1 we have further clarified the purpose of the study in the abstract.

2. we have split the original content into two parts "discussion" and "conclusion", so as to avoid confusion.

Reviewer 2 Report

Relevant research in a little studied field such as adult education.
A data analysis program is used that allows reliable and contrasted results to be obtained.
The manuscript deals with the literature on adult education in online learning. And it is a valuable topic for the scientific field and for society.
It provides some useful conclusions and proposals to think about changes in this field.
To improve the text, it would be necessary to explain a little better how the items analyzed were selected, in the methodology section.

Author Response

Dear Reviewers

        Your comments were very on-target, which was very helpful in improving the paper. We have followed your comments and specifically, we have added “Literature Analysis” to the “Research Method” section to explain why we chose those items for analysis. 

Reviewer 3 Report

This work is very timely. I appreciate the approach you took to methodically analyze the literature. As you noted, limiting to WoS SSCI is a limiting factor in your work. 

Author Response

Dear Reviewers

        Your comments were very on-target. Thank you for your professional and insightful review.It is necessary to include more literature in order to guarantee the scientific validity of the study. However, the editorial board only gave us 5 days to upload the revised manuscript. In this limited time, we have done two remedial efforts so far: 1. we have referenced some valuable studies that were not included in SSCI in the introduction and discussion and conclusion sections, and 2. we have clearly stated the limitations of the study in the paper. In our future work, we will continue to improve our research.

Reviewer 4 Report

The aims of this paper are to examine the factors and related strategies that influence adult online learning and to provide directions for future research based on the status, issues, and trends of existing adult online learning research. The study is very interesting, well presented and thoroughly documented. It is said that in order to ensure the quality of the literature and the credibility of the research results, the authors have chosen articles published in SSCI index journals in Web of Science (WoS) etc. Although, it is a must to do so, however, looking into journals and other sources that are not so well-established or are new, should not be left out from searching. It is also noted, that the issue of sustainable development and SDGs are mentioned but there is not any empirical finding connecting these concepts with the research questions. Connecting sustainability with online learning even on a theoretical basis seems to be essential as it will probably uncover literature gaps. For a further reading see the thematic issue of the Journal of Teacher Education for Sustainability Volume 14, issue 2, 2012.

Author Response

Dear Reviewers       

      Thanks to the clues you gave us, we read a series of papers published in the Journal of Teacher Education for Sustainability, and some of the programmes and practices were very enlightening, such as the M.sc. programme and ICT-based climate change education. We cite some of these papers such as John Huckle, Makrakis et al.

       Thank you for your professional and insightful review, and especially for reminding us to link online learning to sustainable education goals. Since the editorial board only gave us 5 days to revise, I know it is difficult to meet your expectations with the revisions we have made so far, but we are doing our best. We will focus on the topic of online learning and sustainable education in our future research.
